# "Post-GDM support would be really good for mothers": A qualitative interview study exploring how to support a healthy diet and physical activity after gestational diabetes

Rebecca A. Dennison[1]*, Simon J. Griffin[1,2], Juliet A. Usher-Smith[1], Rachel A. Fox[3], Catherine E. Aiken[4,5], Claire L. Meek[6,7,8]

1 The Primary Care Unit, Department of Public Health and Primary Care, School of Clinical Medicine, University of Cambridge, Cambridge, United Kingdom, 2 MRC Epidemiology Unit, Institute of Metabolic Science, School of Clinical Medicine, University of Cambridge, Cambridge, United Kingdom, 3 School of Clinical Medicine, University of Cambridge, Cambridge, United Kingdom, 4 University Department of Obstetrics and Gynaecology, University of Cambridge, NIHR Cambridge Comprehensive Biomedical Research Centre, Cambridge, United Kingdom, 5 Department of Obstetrics and Gynaecology, Rosie Hospital, Cambridge University Hospitals, Cambridge, United Kingdom, 6 Institute of Metabolic Science, Addenbrooke's Hospital, Cambridge, United Kingdom, 7 Department of Clinical Biochemistry, Addenbrooke's Hospital, Cambridge, United Kingdom, 8 Wolfson Diabetes and Endocrinology Clinic, Cambridge University Hospitals, Addenbrooke's Hospital, Cambridge, United Kingdom

* RL423@medschl.cam.ac.uk

**Data Availability Statement:** Anonymized excerpts of the transcripts from the qualitative interviews are reported within the paper. Pseudo-anonymized

## Abstract

### Background

Women with a history of gestational diabetes mellitus (GDM) are at high risk of developing type 2 diabetes mellitus (T2DM). They are therefore recommended to follow a healthy diet and be physically active in order to reduce that risk. However, achieving and maintaining these behaviours in the postpartum period is challenging. This study sought to explore women's views on suggested practical approaches to achieve and maintain a healthy diet and physical activity to reduce T2DM risk.

### Methods

Semi-structured interviews with 20 participants in Cambridgeshire, UK were conducted at three to 48 months after GDM. The participants' current diet and physical activity, intentions for any changes, and views on potential interventions to help manage T2DM risk through these behaviours were discussed. Framework analysis was used to analyse the transcripts. The interview schedule, suggested interventions, and thematic framework were based on a recent systematic review.

### Results

Most of the participants wanted to eat more healthily and be more active. A third of the participants considered that postpartum support for these behaviours would be transformative, a third thought it would be beneficial, and a third did not want additional support. The

transcripts are available via the University of Cambridge Data Repository: https://doi.org/10.17863/CAM.76015. Formal requests for access will be considered via a data sharing agreement that indicates the criteria for data access and conditions for research use and will incorporate privacy and confidentiality standards to ensure data security.

**Funding:** RAD was funded by a PhD studentship from the National Institute for Health Research (NIHR) School for Primary Care Research (SPCR; SPCR-S-S102). This paper presents independent research funded by the NIHR SPCR. The views expressed are those of the author(s) and not necessarily those of the NIHR, the NHS or the Department of Health. JAUS was funded by a Cancer Research UK Cancer Prevention Fellowship (C55650/A21464). SJG is supported by the Medical Research Council (MC_UU_12015/4). The University of Cambridge has received salary support in respect of SJG from the NHS in the East of England through the Clinical Academic Reserve. CEA is supported by an Action Medical Research Grant (GN2778) and a Medical Research Council New Investigator Research Grant (MR/T016701/1). CLM is supported by the Diabetes UK Harry Keen Intermediate Clinical Fellowship (DUK-HKF 17/0005712) and the European Foundation for the Study of Diabetes – Novo Nordisk Foundation Future Leaders' Award (NNF19SA058974). The funders had no role in study design, data collection and analysis, decision to publish, or preparation of the manuscript.

**Competing interests:** The authors have declared that no competing interests exist.

majority agreed that more information about the impact of diet and physical activity on diabetes risk, support to exercise with others, and advice about eating healthily, exercising with a busy schedule, monitoring progress and sustaining changes would facilitate a healthy diet and physical activity. Four other suggested interventions received mixed responses. It would be acceptable for this support to be delivered throughout pregnancy and postpartum through a range of formats. Clinicians were seen to have important roles in giving or signposting to support.

## Conclusions

Many women would appreciate more support to reduce their T2DM risk after GDM and believe that a variety of interventions to integrate changes into their daily lives would help them to sustain healthier lifestyles.

## Introduction

An estimated 17.8 million pregnancies resulting in live births were affected by gestational diabetes mellitus (GDM) worldwide in 2015 [1]. Estimates of prevalence vary greatly within and between regions and countries: the Middle East and North Africa have the highest prevalence at a median 12.9% of pregnancies affected (range 8.4 to 24.5%) and Europe has the lowest prevalence at 5.8% (range 1.8 to 22.3%) [2]; in Eastern and South-Eastern Asia, prevalence is 10.1% (95% confidence interval 6.5% to 15.7%) [3]. Approximately 5% of UK pregnancies were affected in 2015 [4]. Compared to women of an ethnicity associated with a high risk of GDM who are born in a Western country, many women who migrated from their native country to a Western country have higher rates of GDM [5].

GDM is associated with increased risk of pregnancy complications in both mother and baby, and maternal cardiometabolic disorders in later life [6]. Approximately a third of women with GDM are diagnosed with type 2 diabetes mellitus (T2DM) by 15 years postpartum, with recent data suggesting that the increased risk is sustained over time since GDM rather than being limited to the first few years after delivery [7]. T2DM risk factors including high body mass index (BMI) and ethnicity further increase T2DM risk in women who have had GDM: development of T2DM is 18% (95% confidence interval 5–34%) higher per unit BMI at follow-up, and 57% (95% confidence interval 39–70%) lower in White European populations compared to other populations (adjusting for ethnicity and follow-up) [7]. Women from Asia were found to have the highest incidence rate of T2DM after GDM at 46 cases per 1,000 person-years [8]. Factors such as poorer pregnancy glucose tolerance requiring treatment with insulin have been found to further increase risk of T2DM [9]. Overall, women who had GDM are 7–10 times more likely to develop T2DM over their lifetime than women with normoglycaemic pregnancies [7, 10, 11].

In addition to lifelong annual screening for diabetes after pregnancy, women with GDM should be offered postpartum lifestyle advice regarding weight control, diet and exercise [4]. Nevertheless, most women who have had GDM do not attempt or sustain changes to reduce modifiable risk factors but maintain lifestyles that increase their diabetes risk, and many show discrepancy between T2DM risk perception and behaviour [12]. Existing behaviour change interventions have focused on promoting physical activity and a healthy diet, while others have supported breastfeeding after GDM [13]. Intervention modes include group, individual

and remote interventions, or a combination of approaches [13]. Positive effects on preventing T2DM progression are frequently observed but can be limited due to poor engagement, particularly in intensive interventions like the US Diabetes Prevention Programme, in this population [13–16].

In order to understand the facilitators and barriers towards lifestyle in women with a history of GDM, we conducted a qualitative synthesis of their views on reducing their risk of developing diabetes postpartum through lifestyle and behaviour changes [17]. We found that women who had had GDM identified themselves primarily as mothers who prioritised their family above themselves [17]. This motivated some to adopt healthy diets and to be active, but a need for resources, time, energy, information and support prevented others from making changes [17]. From these findings, we developed a set of recommendations for promoting a healthy lifestyle after GDM [17]. Only one of the 21 included studies was set in the UK (interviews with 35 women in total to explore influences on postpartum health behaviours and the feasibility of diabetes prevention intervention [18]) and we are aware of only one other UK study that has been completed more recently (interviews or focus groups with 50 women in total also to explore influences on postpartum health behaviours and preferences for lifestyle support [19]).

There is therefore a gap in recent literature in the UK population surrounding the acceptability of recommendations for intervention after GDM in a real-life context. In this study, we sought to address this gap by exploring the views of women with a history of GDM on possible interventions to support healthy diet and physical activity to reduce diabetes risk, in addition to participants' own suggestions. We aimed to identify the most promising interventions for future development.

## Participants and methods

The 'Diet, Activity and Screening after gestational diabetes: an Interview Study' (DAiSIeS) was approved by the West London and GTAC Research Ethics Committee (reference 19/LO/ 0441).

### Recruitment

Participants were recruited from the Rosie Hospital in Cambridge and Peterborough Hospital. These sites were chosen to provide socioeconomic and ethnic diversity, and represent views from those attending both secondary and tertiary centres offering GDM and obstetric care. Posters were displayed at antenatal clinics to promote awareness of the study. Research midwives identified eligible participants from medical records, and sent them a postal or email invitation and participant information sheet describing the study. Those who were interested in taking part contacted the midwives, and the study researcher (the first author) called them to provide an opportunity to ask questions and arrange the interview.

We planned to interview approximately 20 women in order to reach data saturation, a comparatively large sample size based on the relatively low information power anticipated [20]. This was because this study had a broad aim (to explore the participants' views on potential interventions) and the sample was not very specific (participants had a recent history of GDM but no criteria relating to lifestyle behaviours). One the other hand, the interview schedule and framework analysis were structured around the recommendations made in our systematic review [17], which increases the information power. As widely recommended for data saturation [21], we finished recruitment after several interviews did not lead to novel findings. We interviewed all 20 participants who wanted to take part. Although we did not record the final

number invited, uptake was estimated to be around 50% for women spoken to directly and around 5% for women who were sent a letter or email only.

## Inclusion criteria

Participants were recruited if they had any history of GDM, were over 18 years old, and between 12 weeks and four years postpartum. This timeframe was chosen to allow sufficient time for new mothers to recover from pregnancy and attend postpartum follow-up, and so that all pregnancies were managed according to the National Institute for Health and Care Excellence (NICE) guidelines that were updated in 2015 [4]. NICE recommends screening for GDM with a 75g 2 hour oral glucose tolerance test (OGTT) in women with one or more risk factors (BMI greater than 30 kg/m$^2$, previous baby weighing 4.5 kg or more, previous pregnancy affected by GDM, family history of diabetes, and ethnicity with a high prevalence of diabetes) [4]. Diagnostic cut-offs were defined according to local protocols: at Peterborough Hospital, those with a fasting value ≥5.6 mmol/l or 2 hour value of ≥7.8 mmol/l were diagnosed with GDM (NICE guidelines [4]); at the Rosie Hospital, those with a fasting value ≥5.1 mmol/l, 1 hour value of ≥10.0 mmol/l or 2 hour value of ≥8.5 mmol/l were diagnosed with GDM (International Association of Diabetes in Pregnancy Study Groups (IADPSG) criteria [22]). Screening usually takes place at 24 to 28 weeks gestation, although can be repeated if the clinicians suspect GDM has developed. Following GDM diagnosis, women are closely managed with the aim of reducing glycaemia. This involves blood glucose monitoring, diet and exercise, and sometimes insulin and metformin medication.

Women who had experienced adverse pregnancy outcomes (such as stillbirth, neonatal death or major congenital anomaly), participated in a pregnancy or GDM-related intervention that was in addition to or in place of routine care (such as a clinical trial) or were considered unsuitable for any other reason at the discretion of the midwives who had access to their medical records were not invited.

## Interview process

A single semi-structured interview was conducted at the time and private place of the participants' choice. Children were welcomed in order to facilitate attendance. Firstly, the interviewer (the first author) introduced herself (as a public health PhD student with training in qualitative methods but little interviewing experience) and the purpose of the interview (to listen to their experiences of GDM pregnancy and, particularly, postpartum in order to improve support). Participants then gave written informed consent, confirming that they understood the purpose and procedure of the study and that they could stop the interview at any point, and were happy for it to be audio-recorded.

Our previous systematic review informed the interview guide and suggestion cards, which were adapted from 20 recommendations (Fig 1 and S1 Table) [17]. The first part of the interview focused on diet and physical activity while the second part focused on screening for T2DM (reported separately [23]); ten suggestion cards were used as prompts in each part. We conducted three pilot interviews and collected written feedback from our patient and public involvement group, composed of mothers with GDM. This feedback was incorporated into the final version, which was refined after reflection on the first interviews. This involved changing the first question of the interview schedule from asking about their current diet (which we expected would be an easy discussion topic) to asking them to describe "what your GDM pregnancy was like for you". Although this was not the focus of the interview and sometimes brought up upsetting accounts, the participants appreciated being able to tell their 'story' and it proved to be useful context for the remainder of the interview. Additionally, more

**Systematic review recommendation**

**Suggestion card used in interviews**

*Role of mother and priorities*

1. Highlight the benefits to the family of the mother being healthier and role modelling healthy lifestyle to children as the incentive for change, alongside preventing diabetes (M)

2. Include childcare in face-to-face interventions, if the sessions are not for children (M)

*Support from family and friends*

3. Promote healthier lifestyles in the wider family and friends (M)

4. Encourage the wider family and friends to promote healthy lifestyles in mothers and support them practically (such as relieving housework burdens) (H)

5. Include the family in interventions (e.g. information for partners and children) (M)

6. Encourage and facilitate women to exercise with others/a buddy (M)

*Demands of life*

7. Provide guidance about how to buy and prepare healthy, tasty food efficiently (H)

8. Provide guidance about exercising around the house and as part of daily routines (M)

*Personal preferences and experiences*

9. Support women to maintain healthy behaviour/diet in challenging situations – e.g. social gatherings, breastfeeding, at work (particularly for vulnerable groups) (L)

10. Highlight the wider benefits of healthier lifestyle (such as reducing stress and weight as well as diabetes risk) (H)

*Knowledge and information*

11. Make information, resources and training easily accessible and make interventions available to start immediately after pregnancy (or during pregnancy) (H)

12. Ensure that interventions are culturally appropriate and recommendations allow maintenance of women's identity (H)

13. Ensure that care providers consider women's attitude towards diabetes and advise them on their risk appropriately (L)

14. Promote a long-term perspective about maintaining healthy lifestyle, with an 'every little helps' approach, and include the importance of both diet and activity (M)

*Finances and resources*

15. Provide information about low-cost or money-saving healthy behaviours and resources; interventions should be free (H)

*Format of intervention and other*

16. Recommend increasing fruit and vegetable intake, reducing sugar and substituting with healthier ingredients or methods to improve diet (M)

17. Recommend flexible exercise such as walking and those performed around the home or with the baby to increase physical activity (rather than attending gyms or classes) (H)

18. Ensure interventions have web-based components but encourage additional face-to-face contact (they should not depend on women attending sessions) (L)

19. Deliver and promote interventions from recognised/trusted sources (e.g. the healthcare provider or a dietitian) (L)

20. Promote establishment of systems to monitor progress and accountability (through an intervention or ensure the participant establishes this themselves) (H)

1. More information about diet/ exercise on T2DM risk

2. More information about diet/ exercise on wider health

3. More information about diet/ exercise on family

4. Suggestions for healthy families

5. Support to exercise with others

6. Advice about how to eat healthily

7. Advice about exercising with a busy schedule

8. Advice about sustaining changes

9. Advice about saving money

10. Advice about monitoring progress

**Fig 1. Adaptation of recommendations developed in the qualitative synthesis [17] to the DAiSIeS interview schedule.** H: high confidence; M: medium confidence; L: low confidence in the recommendation in accordance with the GRADE-CERQual evaluation [24].

signposting was incorporated into the interview such as "to help us understand any lasting impact GDM might have had. . ." or "before we talk about exercise, I'd like to ask you about your diet. . .".

Participants were first asked to share their experience of GDM in order to understand the background for their behaviours and attitudes. They then described their current dietary and physical activity habits and perceptions of the influence of GDM. We discussed whether any support for these behaviours would be helpful, and what format might be most effective (such as online, face-to-face with clinicians, peer support groups, etc.). Participants were asked about their own ideas first, then to comment on the ten suggestion cards provided by the interviewer (e.g. whether they agreed, disagreed or would add anything). It was emphasised that disagreement (they probably would not find that intervention helpful) as equally useful to hear as agreement (they might find that intervention helpful). If the participants did not want to make

any further changes themselves, they were asked what they thought might help others with GDM based on their own experience. Prompts were used as required. Occasionally the interviewer decided not to show a particular participant a suggestion card if she considered that it was not appropriate given the earlier content of the interview and sometimes the participant did not provide clear feedback (e.g. due to natural distractions or initiating a different train of thought). We then discussed how to facilitate attendance at diabetes screening after GDM in the second part of the interview [23]. Finally, demographic information (including age band, ethnic group, employment status and pregnancy history) and interview feedback were collected through a short questionnaire. The interviewer recorded reflexive field notes on each interview including the context (e.g. setting, if anyone else was nearby) and subjective reflections.

## Analysis

A professional transcription service transcribed the interview recordings. The first author checked the transcripts for accuracy and removed names, places and other potentially identifiable information.

We used a framework approach to analyse the interviews [25], with the aid of NVivo 12 (QSR International Pty Ltd; version 12; released 2018):

1. Familiarisation: We familiarised ourselves with the data by listening to the recordings and reading the transcripts and field notes, and making notes about important concepts.

2. Identifying a thematic framework: We developed a thematic framework that was based on the suggestion cards then refined it as required upon analysing the interviews by incorporating additional repeating concepts (such as communication requires a positive, non-judgemental tone). We distinguished between suggestions initiated by participants and responses to the suggestion cards in the analysis. The final codebook for the framework is reported in S2 Table.

3. Indexing: Next, we coded each transcript according to the thematic framework.

4. Charting: We drew charts to summarise what each participant said in relation to each part of the thematic framework. One row was used for each participant interviewed, and one column was used for each code within the framework.

5. Mapping and interpretation: We carefully studied the charts for repeating or characteristic ideas to describe and explain the phenomena observed. Where differences and deviant cases were observed, we attempted to understand the ways in which they different and why, according to the information the participants provided.

The first author coded all of the transcripts and developed the charts, and another author coded and charted four transcripts to ensure general agreement with the coding strategy and discuss alternative explanations. The other authors read some or all of the transcripts and charts in order to support interpretation. During these discussions, we considered the authors' clinical (obstetrics, diabetes and general practice) and non-clinical backgrounds and made notes to record the analytical and interpretational decisions.

To supplement the qualitative analysis, we classified the participants' collective response to each suggestion card as overall agreement, disagreement or mixed in order to create a general indication or impression of their views. The classification was based on the authors' interpretations of the participants' responses: whilst we counted the number of agreements or disagreements, we also considered the vigour with which each participant responded. Where the classification was not obvious, consensus among the authors was sought.

We also invited the participants to provide feedback on a summary of the findings (not the transcripts) and incorporated any responses into the final version.

## Results

Twenty participants were interviewed between June 2019 and February 2020; 11 were recruited from Peterborough Hospital and nine from the Rosie Hospital in Cambridge. Most interviews took place in homes and two were at a hospital. The median (interquartile range) number of pregnancies per participant was 2 (1–2.25), with 1 (1–2) pregnancy affected by GDM. None of the 16 participants who had had a diabetes screening test since pregnancy had been diagnosed with T2DM. Table 1 shows the participants' characteristics at the time of the interview. Interviews had a mean duration of 38 minutes (range 21–62 minutes).

Many participants had made significant lifestyle changes during pregnancy to manage GDM, and felt as if GDM had ruined their pregnancy or their lives had revolved around their blood glucose levels. The perception of care they received during pregnancy was mostly very good, yet several mentioned not wanting to have another child in fear of GDM.

Overall, the participants were eager to make changes to and take responsibility for their health. Many highlighted the importance of their individual mindset and desire to be helped. Seven participants had sufficient knowledge about healthy diet and exercise going forward, or knew where to find more support if they needed it. Seven participants acknowledged that more postpartum follow-up would be helpful, but they had been able to manage. The remaining participants reported sentiments such as "*I don't feel like I've been given the help that I think there should be really out there*" [Participant 1, attempting healthier postpartum lifestyle but felt unsupported overall], "*post-GDM support would be really good for mothers*" [P2, healthier, unsupported] and two participants explained that they had been unaware of an association between GDM and T2DM. Those who had struggled through pregnancy found the postpartum period particularly challenging.

The participants' views on suggestions to support a healthy postpartum diet and physical activity are summarised in Table 2, plus S3 Table indicates their agreement with each suggestion card. Many of the participants were positive towards the suggestions despite already making healthy changes. Others had mixed responses because they had specific questions, and one participant felt that "*they're [the suggestion cards] all quite similar, aren't they? You know, I think I know those things already. . .*" [P3, not attempting healthier lifestyle].

### Information and understanding (suggestion cards 1 and 2)

Most of the participants felt that they would benefit from more information about the impact of healthy behaviours on their diabetes risk. Some would add this to existing knowledge, whereas others had poor awareness of the long-term implications of GDM because they hadn't been told or remembered. It was important that information was adapted to mothers who had had GDM and perceived themselves to be knowledgeable ("*not sort of trivial, such as 'eat a healthy balanced diet, exercise more'*" [P6, healthier, supported], and focused on how to be healthy in relation to T2DM.

Opinions varied about information on the impact of healthy diet and exercise on their wider health since most already had general awareness or found that this was covered by existing postpartum support, such as that provided by children's centres.

### Improving diet (suggestion card 6)

The majority of the participants were attempting to eat healthily by continuing elements of their GDM diet. Further guidance or tips would help them to do this because they received little or no information about what to eat after delivery (in contrast to pregnancy).

**Table 1. Participant characteristics at the time of the interview.**

|  | N (percent) |
|---|---|
| **Age band** |  |
| 26–30 years | 3 (15) |
| 31–35 years | 9 (45) |
| 36–40 years | 6 (30) |
| ≥41 years | 2 (10) |
| **Ethnicity** |  |
| White British or European | 14 (70) |
| Asian or Asian British | 6 (30) |
| Chinese | 2 (10) |
| Indian | 3 (15) |
| Any other Asian background | 1 (5) |
| **Education level** |  |
| Secondary or further (GCSEs, A levels, BTEC, apprenticeships or equivalent) | 5 (25) |
| Higher (Bachelor's degree or equivalent) | 6 (30) |
| Postgraduate (Master's degree, PhD or equivalent) | 9 (45) |
| **Employment (when not on maternity leave)** |  |
| Full time | 10 (50) |
| Part time | 9 (45) |
| Home parent | 1 (5) |
| **On maternity leave** |  |
| Yes | 11 (55) |
| No | 8 (40) |
| NA | 1 (5) |
| **Household status** |  |
| Lives with partner | 18 (90) |
| Does not live with partner | 2 (10) |
| **Number of children** |  |
| 1 | 6 (30) |
| 2 | 9 (45) |
| ≥3 | 5 (25) |
| **Number of pregnancies affected by GDM** |  |
| All pregnancies affected by GDM | 13 (65) |
| Have also had normoglycaemic pregnancies | 7 (35) |
| **Management of GDM** |  |
| Required medication (metformin and/or insulin) | 10 (50) |
| Managed by dietary and lifestyle changes alone | 10 (50) |
| **Experience of GDM pregnancy and postpartum[a]** |  |
| GDM management required significant/challenging lifestyle changes | 17 (85) |
| They were attempting to maintain a healthy postpartum lifestyle | 14 (70) |
| They felt adequately supported to maintain a healthy postpartum lifestyle | 10 (50) |

[a]Elicited from transcripts. NA: not applicable.

A couple of participants were uncertain because the GDM diet wasn't a 'normal' healthy diet, such as eating peanut butter instead of fruit. Others wanted advice that was relevant to other aspects of their new situation, including managing cravings, balancing healthy diet with calorie intake for breastfeeding, with children of different ages, and different family mealtimes.

**Table 2. Summary of the themes and participants' agreement with whether the suggestion cards will support healthy diet and physical activity.**

| Theme | Overall response | Illustrative quotations |
|---|---|---|
| Information and understanding | Suggestion card 1: agree<br><br>Card 2: mixed | • "I think the more information a person can have, the more able they are to make an informed decision, and I think that's, especially as a mum, what you want." [P4, not healthier]<br>• "I think people know about healthy diet and exercising and they know that's good for you and good for your weight but whether people can do it might be another thing." [P2, healthier, unsupported]<br>• "I don't think that's [card 2] as necessary, because I feel like that's widely available, and I know that. But in terms of the link to diabetes [card 1], I didn't know that." [P5, not healthier] |
| Improving diet | Card 6: agree | • "The diet I was given to follow during pregnancy, bits of it felt very counter to what I understood to be healthy... I understood for the purposes of really stabilising my blood sugar that was important to do but... My vision of what a healthy long-term diet are don't include most of those features... I suppose that would be quite useful if there was some sort of follow-up information, 'Okay, you've done this, now you're going to rebound a bit and we're not asking you to keep it like this but it would be a good idea to...', you know, 'These ones are worth following, these ones aren't.' Maybe that exists but I don't think I've seen it." [P6, healthier, supported]<br>• "If you have a clean track of what you want to eat and what are the things that add up your calories and what other things are good for you to control your diabetes, like the sugar levels. I think that can help a lot." [P7, healthier, supported] |
| Improving physical activity | Card 5: agree<br><br>Card 7: agree | • "Like how to exercise around the home, because it's really difficult trying to work out when you're going to fit everything in, especially when you've got a small person that generates more washing than you could ever imagine..." [P8, not healthier]<br>• "Having a baby carrier... you can keep an eye on them and they are happy because they're [across your chest]. But also it gives you both your hands free to do stuff. Also it is exercise because you're carrying them around and they're getting heavier and heavier. Just make sure you get a good one that supports your back." [P9, healthier, supported]<br>• "Just a bit of a pointer in where to go and who to go to and what also would fit into a family life in terms of finance and childcare, and potentially meeting up with other mums or other people who've had diabetes as well." [P10, healthier, supported] |
| Family | Card 3: mixed<br><br>Card 4: mixed | • "...Sometimes [my children] won't agree to what you give... there's green food–'I don't want', they want some kind of pizza or burger all those things but still I somehow try to convince her with this kind of food." [P7, healthier, supported]<br>• "We both [her and her husband] did a lot of research... we are much more health-conscious and we try to exercise more, so I think we've both changed our lifestyle, and it carries on as well." [P11, healthier, unsupported]<br>• The children "don't struggle with blood sugars, they don't struggle with not being able to get out and get fresh air." [P10, healthier, supported] |
| Money | Card 9: mixed | • "I mean it's always good to know about how to save money but I just don't think people don't go on a healthy diet because of money problems." [P2, healthier, unsupported]<br>• "I don't think there's much useful guidance about maintaining that kind of healthy diabetes-friendly diet on a budget actually. A lot of healthy meals tend to be focused on things like lasagnes and stuff like that, like big batch cook things that aren't necessarily the right thing for someone who is trying to like minimise diabetes risk to be eating." [P12, healthier, unsupported] |

*(Continued)*

**Table 2.** (Continued)

| Theme | Overall response | Illustrative quotations |
|---|---|---|
| Monitoring | Card 10: agree | • "It's always nice to see your results to see some sort of benefits that you've been achieving, I think spurs you on." [P13, healthier, supported]<br>• "Apart from contacting my doctor to get a HbA$_{1c}$ test every year, no one's contacted me to say, 'Have you made any lifestyle changes, how you getting on?' So, it's almost like you're just left to get on then afterwards." [P14, healthier, supported] |
| Sustainability | Card 8: agree | • "I think that would be really useful because I know a lot of people would perhaps make the change and then slip back into bad habits." [P15, healthier, unsupported]<br>• "Because people lose motivation quite quickly, they have the best intentions, and then. . . I think that's probably where support groups that motivate one another would help." [P14, healthier, supported] |
| Delivery of support or interventions | NA | • "I think it needs to be someone that's personable, because I think from my experience, sometimes when you go to the hospital you get really nice consultants and sometimes you don't. . . just needs to be someone that can be relatable and friendly and isn't going to come across hostile or judgey, it is just here if you need a chat sort of thing." [P16, healthier, unsupported]<br>• "That is all while you are pregnant but then maybe afterwards you don't get that side to carry on. . . Like a little leaving parcel of like here's a little pack of how to keep going with the good work you've done, and help you prevent it in the future and just make it clear that actually although it is gestational and it goes, it doesn't mean you are rid forever." [P16, healthier, unsupported]<br>• "If somebody had said to me at that point, 'You need to be eating this, this and this,' I think I'd have probably cried", and "I just think that rather than checking, so that people don't feel like they're being checked up on, because if you've been ill and you haven't got out of the house, you don't want to feel like you're failing your child." [P8, not healthier]<br>• "Especially when you're doing feedings. . . late night feeds or whatever, you can sit and have a look at your phone and get that support 24/7." [P14, healthier, supported] |

Overall agreement is based on the authors' interpretation of the responses. Not all participants were shown each card, and some did not comment or agreement was unclear. For each quote, we report the participant number, whether they were attempting healthier postpartum lifestyle or not (healthier/not healthier), and whether overall they felt supported to do this (supported/unsupported).

NA: not appropriate.

It was important for advice to be individualised ("*how to keep your diet. . . right for you*" [P1, healthier, unsupported]) and in accordance with their palate or culture.

Two participants thought that they had the necessary knowledge but that other people did not. Three participants already had enough information by drawing on previous experiences and GDM diets.

## Improving physical activity (suggestion cards 5 and 7)

Although many participants reported doing less physical activity than before pregnancy, several prioritised running (while their partner looked after the children), dance classes or home workouts. Others did lower-intensity activity, like pushing the buggy. Many wanted to do more exercise, and felt this would be achievable when the children were older, they finished breastfeeding, or had better recovered from pregnancy.

Help to exercise with others was frequently considered to facilitate physical activity because it had helped them in the past or they walked with others now, or it might make exercise less boring. Some preferred mother-and-baby classes or GDM groups, which would be accessible, and could provide an opportunity for socialising and sharing experiences alongside exercise. Local groups might need signposting because they didn't know where to find them or hadn't thought to look. Conversely, a few felt distracted when exercising with others or liked to exercise at their own pace.

Almost all the participants were eager for advice about how to be active alongside a busy schedule (including around the home and exercise for the whole family together), explaining that was the thing they had issues with and hadn't received any advice about. Appropriateness for postpartum period was important: one participant suggested cards with postpartum-friendly exercises "*like little diagrams and exercise routine that build the further on you get in your health. . . especially to what kind of birth you've had*" [P16, healthier, unsupported]. Several participants shared what had helped them, including splitting exercise throughout the day and using a baby carrier.

### Family (suggestion cards 3 and 4)

A young family made having a healthy lifestyle harder than it used to be due to increased demands on their time, and the need to meet others' dietary preferences/requirements. On the other hand, parenthood could provide new opportunities: one participant's older child encouraged her to exercise, and others walked with their antenatal groups. Some also found that their children motivated them to be healthier because they wanted to stay well for their family, prevent unhealthy habits in their children, and/or their partners wanted to be healthier too after they both learnt more about diabetes.

They therefore had mixed views regarding whether more information about the impact of healthy diet and exercise on their family would help them. Some participants reasoned that being healthy was something they would do as a family whereas others felt that it was only relevant to themselves. Others already knew the information, or it had been provided by their health visitor (although not everyone received this kind of guidance). Similarly, the suggestion of ways for the family to be healthier received mixed agreement; those that agreed wanted practical tips for fitting a healthy lifestyle in with family life, ideas for activities involving wider family and friends, and how to easily adapt child-friendly recipes for parents.

### Money (suggestion card 9)

Twelve participants were in favour of advice about saving money and maintaining a healthy lifestyle because they found generic advice could not be applied to diabetes prevention. They also needed healthy options for the family to do, particularly because costs increased with a larger/growing family. Conversely, other participants considered that cost didn't prevent a healthy lifestyle because cheap or free options were available. Some noted that cooking from scratch was already cheaper than buying prepared food; they therefore had fewer options for saving more money.

### Monitoring (suggestion card 10)

Almost all of the participants had positive views towards monitoring their progress after pregnancy, anticipating it to make a big difference. They discussed either monitoring themselves (by recording their weight, diet, exercise, calories in and out, or 'nice' things like visiting the park) or through meeting with a health professional. Importantly, it was seen as a way to maintain motivation for changes through seeing their achievements and the benefits, or recreate targets like they had during pregnancy. A health professional could give more information and

feedback on individual diabetes risk and blood glucose control. At the same time, several were cautious that monitoring could have the opposite effect: seeing weight increase could be demoralising and involving others might be stressful.

## Sustainability (suggestion card 8)

With the exception of those who were not attempting to eat a healthier diet and exercise more, the participants wanted advice about sustaining changes and knew that maintaining a healthy lifestyle would be challenging. In practice, they felt that sustainability could be facilitated through the earlier themes; for example, that advice about healthy food that is suitable for the whole family, exercises that can be done around the house, and more follow-up will all help them to maintain behaviours to reduce their risk of T2DM.

## Delivery of support or interventions

The participants also suggested how the above support could be delivered. This included the preferred format (including in-person peer support groups, appointments with healthcare professionals and written information), source and timing.

**In-person peer groups.** Seven participants wanted to share experiences in a peer support group throughout GDM pregnancy and postpartum. "*Mum-centric*" postpartum groups [P13, healthier, supported] could include tips for reducing diabetes risk, be linked to exercise classes and hosted through children's centres, where other educational classes, such as for breastfeeding and postpartum mental health, already took place.

**Appointments with healthcare professionals.** This was the most frequently suggested intervention. Midwives, hospital diabetes teams, health visitors and GPs had provided GDM care and were a trustworthy and respected source of information.

Most participants were keen to receive advice about postpartum diet and exercise, and long-term diabetes risk during pregnancy. It would be good to be briefed while they were most aware of GDM, knowing that more information would follow. Only one participant felt that this would overwhelm her because there was already too much to think about during pregnancy.

Similarly, four participants felt that follow-up should be mentioned, in a casual way, while they were on the maternity ward or alongside other discharge information. Women who had more complicated births spent more time in hospital and generally felt abandoned with regards to GDM at that time, therefore would like the opportunity to make sure that they "*knew the plan of action*" with a professional [P4, not healthier]. One participant disagreed because she lost all of the many discharge papers she was given.

Thirteen participants discussed attending postpartum appointments with a clinician. Many suggested that GDM follow-up become part of the six-week mother-and-baby healthcheck with the GP, which would be after the initial overwhelming stage. In practice, this appointment focused on the baby, which was very important, but they too needed time with an expert to debrief: to be asked how they were, have some reassurance, discuss what to do next and, notably, receive feedback on each blood test result.

**Written information.** The participants thought that written information about postpartum lifestyle would be beneficial, such as a booklet, website or interactive smartphone app, like they had sought during pregnancy. Support would then be available at all times, including during night feeds. One proposed a "*website that can make suggestions or to have a community of people with GDM who share recipes, what their concerns are*" [P11, healthier, unsupported], because social media groups had the potential to be informative and supportive. Regardless of format, this would be most beneficial if it was provided alongside face-to-face care or if a clinician directed them to trusted resources.

**Delivery of messages.** Six participants, particularly those with specific struggles during pregnancy and/or postpartum, felt strongly that information should be shared in an individualised and sensitive fashion. Positive framing was important in the context of postpartum stress, diabetes-related fear, and outstanding feelings of guilt or judgment from having GDM.

## Discussion

In this study, we explored the views of 20 mothers with recent GDM towards suggested support for having a healthy diet and being active in light of their T2DM risk. These women thought that additional advice about how to eat healthily and exercise when they were busy, and practical suggestions for making these changes sustainable in their context, would most help them to reduce their risk of T2DM. Many wanted more individualised information about their long-term risk of T2DM after GDM, and how they might mitigate that risk, but they often knew enough about the overall benefits of a healthy lifestyle. Although written information in any format would be acceptable, access to other mothers with GDM and a clinician talking to them about follow-up in a supportive manner was anticipated to be beneficial.

The DAiSIeS study was designed to build on our recent qualitative synthesis [17], bridging the gap between barriers and facilitators to diabetes prevention behaviours and intervention programme design. Comparing the findings of that review [17] and this interview study, we found that influences on healthy diet and exercise were similar, such as spending time with children instead of exercising and how the family could facilitate healthy behaviours. We had reported a lack of time and energy as barriers to healthy lifestyles; this was also true in this interview study, particularly in the early postpartum period that was considered to be a time for learning to adapt to life with their new baby. Women were more supportive of integrating activity into their daily routine than of participating in family-based exercise activities, which had appeared to be important in the review. Even though most of the DAiSIeS participants had a positive experience of GDM pregnancy and knew about having a healthy lifestyle, many felt that more specific information about lifestyle behaviours in T2DM risk prevention was important (such as what foods would be best for them to eat). This echoes a participant in Lindmark *et al.* 2010 who said '. . .even if it is old knowledge it is good to hear it once more' [26]. The participants tended only to maintain selected elements of their GDM diet, which aimed to minimise spikes of high blood glucose during pregnancy, because they considered it was too extreme to sustain (such as a slight/moderate reduction in carbohydrate intake and strict avoidance of all high glycaemic index foods and high sugar fruit). Educational interventions may therefore support them to learn what things to continue and what not to in order to lose weight and maintain a balanced diet. Of particular note, these changes were anticipated help women maintain healthy lifestyle in the long term.

Previous studies have reported varying views regarding the best timing for intervention: some suggest during pregnancy [17, 27, 28] while others suggest postpartum [17–19]. We concluded that women with GDM should be prepared for more specific follow-up interventions such as those described above during their pregnancy, provided that this is done in a sensitive manner, echoing the findings of Ingstrup *et al.* regarding the importance of rapport with peer councillors [29]. In general, any healthcare professional involved in the care of women with GDM can promote a longer-term perspective.

### Strengths and limitations

We used qualitative semi-structured interviews to understand the views of women with GDM towards improving postpartum support for heathy behaviours to reduce T2DM risk. While focusing on their own views towards effective interventions, we based the study design,

interview schedule and analysis on recent systematic review evidence meaning that a clear evaluation of suggestions could be elicited.

The participants were from a mixture of ethnic and socioeconomic backgrounds, although many were more highly educated than the rest of the UK [30]. Nonetheless, the need for post-partum support–therefore the anticipated benefits of interventions–was high in this population, and may be higher in other settings. This may have been influenced by recruitment bias, with more health-conscious women or those in need of particular support more likely to engage in the study. We were also unable to capture the number or any characteristics about the women who were aware of the study (through seeing the posters displayed at GDM clinics or being invited specifically) and chose not to take part. However, the participants' characteristics were comparable to those of the women attending these clinics [31, 32]. We did not collect data on BMI or whether participants were overweight because we did not access their medical record, and to ask them to self-report this could be insensitive during the postpartum period. It is important to facilitate all women to maintain a healthy diet and be physically active after GDM, regardless of their BMI, yet we were unable to comment on whether BMI affected requirements for support. Furthermore, social desirability bias may have led some participants to agree with the prompted suggestions, although it was clarified in the question that negative responses would be as informative as positive ones. Finally, as is true for all qualitative research, other interpretations of the data collected in this study might be possible, although no participants disagreed with the summary of the findings that we sent to them.

## Implications for practice

In this study, the participants were keen to have a healthier diet and increase their physical activity after pregnancy. Importantly, many recognised the dedication and support they would need to sustain changes they had managed during pregnancy. Because intention and self-efficacy, influenced by past experience, have been associated with healthy diet and exercise at one and two years postpartum [33, 34] nurturing these beliefs is imperative. We identified a wide

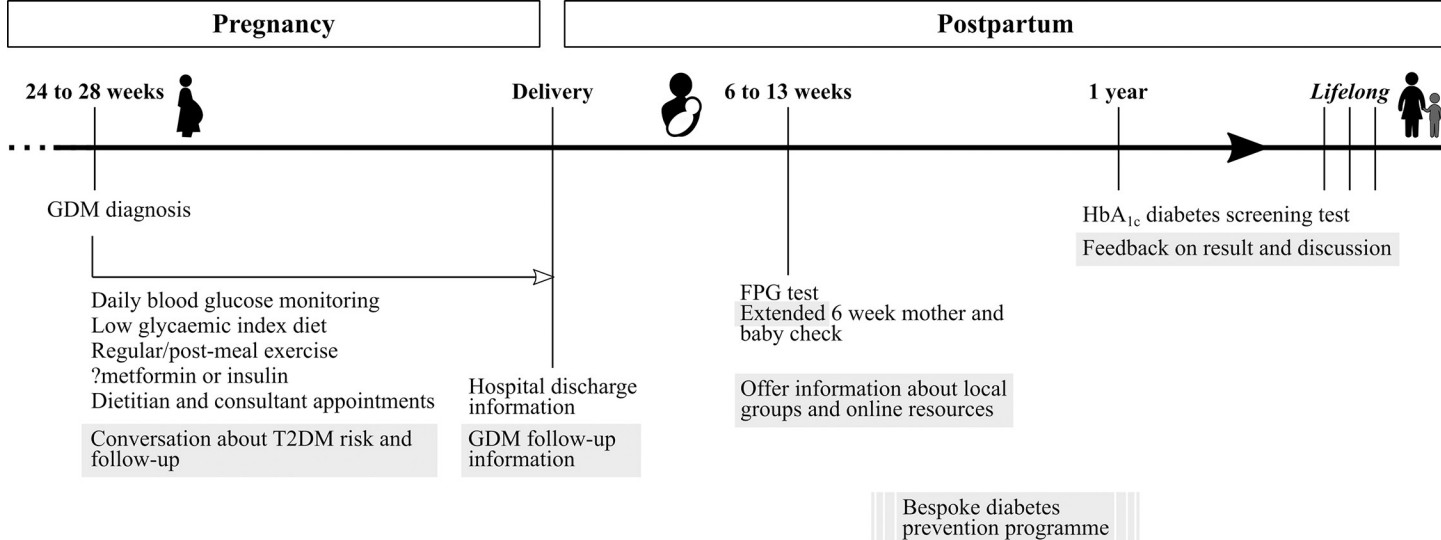

**Fig 2. Summary of key proposed amendments to current GDM pregnancy and postpartum care.** Proposed amendments are shaded in grey. Abbreviations: FPG–fasting plasma glucose test; HbA$_{1c}$ –glycated haemoglobin test.

range of more specific requirements that could be addressed through various multi-faceted approaches (Fig 2).

Our findings support the important role that clinicians play in promoting healthy behaviours and signposting resources during pregnancy and postpartum [35]. Additionally, we suggest that it would be acceptable for the longer-term implications of GDM to be discussed in an informal manner throughout pregnancy and mentioned while mothers are on the postnatal ward. Many studies have reported pregnancy to be a 'teachable moment' for a range of behaviours due to increased motivation and regular contact with health professionals [36, 37]. As a result, informing women about postpartum recommendations in pregnancy is likely to be beneficial although not an end in itself [14].

The participants also expressed interest in a postpartum follow-up appointment. If the blood test was undertaken in advance, the six-week mother-and-baby healthcheck [38] could be extended to include specific GDM follow-up, such as discussion of future plans for diet and exercise going forward in light of the test result. This would also provide an opportunity to ask specific outstanding questions. Since half of mothers receive inadequate time to discuss their own mental and physical health [39], both the mother and GP should have aligned expectations about this appointment.

Postpartum contact also provides opportunity for a healthcare professional to signpost mothers to existing resources. Although some DAiSIeS participants preferred to meet other mothers with GDM, many expressed similar experiences and needs as women without GDM, therefore general postpartum dietary information or exercise classes could be beneficial. A recent study of mothers in a similar area identified the need to increase capability for exercise through signposting to suitable mother-and-baby exercise classes (which would be an environment where they felt comfortable about themselves and bringing their baby), and guidance about how to exercise safely after the birth [40].

Like many others, the women in our study reported accessing and interacting with websites, forums, social media and other sources of written information during pregnancy and postpartum. For example, mothers reported accessing Facebook more frequently in the postpartum period [41], such as to connect with other breastfeeding mothers for advice [42, 43]. Information was accessible at all times and could be informative and supportive, but many users raise doubts about trustworthiness [42, 44, 45]. A recent analysis of posts on Mumsnet and Netmums forums concluded that the support provided does not encourage T2DM prevention because diabetes risk was rarely discussed and users downplayed the seriousness of GDM and its association with lifestyle behaviours [46]. Instead of searching for such groups themselves, mothers could be directed to reliable resources by a trusted professional or body.

## Conclusions

Many women wanted more support to sustain healthy lifestyles to reduce their T2DM risk after a GDM-affected pregnancy. We identified a broad range of interventions that could offer this support. These mothers thought that additional advice about how to eat healthily and be active when they were busy, and tips for maintaining these changes, would help them most. Many wanted more specific information about their long-term T2DM risk, but they often knew enough about the universal benefits of a healthy lifestyle. This support could be provided throughout pregnancy and postpartum, in a range of formats including face-to-face with healthcare providers or peers and online or physical written information. Directing women to existing trusted resources or groups, or adapting existing interventions to the needs of this population is likely to improve care for mothers after GDM.

## Supporting information

**S1 Table. DAiSIeS interview schedule.**
(PDF)

**S2 Table. Thematic framework used to analyse the DAiSIeS interviews.**
(PDF)

**S3 Table. DAiSIeS participants' agreement with whether the suggestion cards will support healthy diet and physical activity.** Agreement was based on the authors' interpretation of their responses. Not all participants were shown each card, and some did not comment or agreement was unclear. Dark green: strongly agree; green: agree; red: disagree; dark red: strongly disagree; grey: not shown or agreement is unclear. A: overall agreement; M: overall mixed response; D: overall disagreement.
(PDF)

## Acknowledgments

We thank the members of our PPI group (GDM Voices) who were involved in the design of the study, the multidisciplinary diabetes in pregnancy clinical teams and research nurses at the Rosie Hospital and Peterborough Hospital for recruiting the study participants, and the participants themselves for giving their time to share their views with us.

## Author Contributions

**Conceptualization:** Rebecca A. Dennison, Simon J. Griffin, Juliet A. Usher-Smith, Catherine E. Aiken, Claire L. Meek.

**Data curation:** Rebecca A. Dennison, Claire L. Meek.

**Formal analysis:** Rebecca A. Dennison, Simon J. Griffin, Juliet A. Usher-Smith, Rachel A. Fox, Catherine E. Aiken, Claire L. Meek.

**Investigation:** Rebecca A. Dennison.

**Methodology:** Rebecca A. Dennison, Simon J. Griffin, Juliet A. Usher-Smith, Catherine E. Aiken.

**Project administration:** Rebecca A. Dennison, Claire L. Meek.

**Resources:** Rebecca A. Dennison.

**Software:** Rebecca A. Dennison.

**Supervision:** Simon J. Griffin, Juliet A. Usher-Smith.

**Visualization:** Rebecca A. Dennison.

**Writing – original draft:** Rebecca A. Dennison.

**Writing – review & editing:** Rebecca A. Dennison, Simon J. Griffin, Juliet A. Usher-Smith, Rachel A. Fox, Catherine E. Aiken, Claire L. Meek.

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
