## [Decision Letter · Decision Letter 0]

5 Mar 2021

PONE-D-21-01829

“Post-GDM support would be really good for mothers”: a qualitative interview study exploring how to support a healthy diet and physical activity after gestational diabetes

PLOS ONE

Dear Dr. Dennison,

Thank you for submitting your manuscript to PLOS ONE. After careful consideration, we have decided that your manuscript does not meet our criteria for publication and must therefore be rejected.

I am sorry that we cannot be more positive on this occasion, but hope that you appreciate the reasons for this decision.

Yours sincerely,

Wing Hung Tam

Academic Editor

PLOS ONE

Additional Editor Comments (if provided):

Thank you for the submission to Plos One. Given the reviewers comment and the small number of highly selected participants on a qualitative review, the manuscript is not considered to be suitable for publication in Plos One.

Reviewers' comments:

Reviewer's Responses to Questions

**Comments to the Author**

1. Is the manuscript technically sound, and do the data support the conclusions?

Reviewer #1: Yes

Reviewer #2: Partly

2. Has the statistical analysis been performed appropriately and rigorously? 

Reviewer #1: Yes

Reviewer #2: No

3. Have the authors made all data underlying the findings in their manuscript fully available?

Reviewer #1: No

Reviewer #2: No

4. Is the manuscript presented in an intelligible fashion and written in standard English?

Reviewer #1: Yes

Reviewer #2: Yes

5. Review Comments to the Author

Reviewer #1: Thank you for the opportunity to review this manuscript. This study aims to conduct a qualitative interview study on the views of women with history of GDM on the need for post-partum support. Given the high prevalence of GDM (affecting 10-20% of all pregnancies in many parts of the world), and the high risk of progressing to T2D after GDM (approximately 7-8x risk compared to non-GDM women), locally-relevant strategies to support women with GDM to reduce progression to T2D are very important. This study provides a systematic assessment of the attitudes and needs of women with GDM, and provide some useful information, in particular towards developing interventions and support networks within the UK healthcare setting and beyond.

Major comments

Introduction

More reference to the risk of T2DM after GDM might be helpful in the introduction. This may include data from systematic review highlighting the increased risk of T2D afer GDM compared to non-GDM pregnancies.

Page. 4 Line 82

It was stated that all women had history of GDM. Please clarify the prevailing diagnostic criteria which would have been used to diagnose GDM in these subjects

Page 6 Line 135

It was stated that none of the participants had been diagnosed with T2D. Please clarify if all the women with GDM had undergone postpartum OGTT screening

Table 1

Is any information available on the proportion of women who were overweight/obese, or their mean BMI?

Page 8 Line 151

Although I understand this is a qualitative interview study, are the authors able to state the number of women who were aware of the link between GDM and T2D, which would significantly affect the interpretation of the results?

Page 14 Line 246-248

Please provide further details on the responses from the participants on the preferred format, source and timing of providing support . Are the responses summarized by the later comments on In person peer groups, appointments with healthcare professionals and written messages?

Reviewer #2: The authors conducted semi-structured interviews with 20 participants with a history of GDM to explored the views of women on possible interventions to support healthy diet and physical activity to reduce diabetes risk and aimed to identify the most promising interventions for future development. The sample size was small and less representative. I also have some other concerns about the interpretations and discussions.

Major comments:

1. The participants were women who were interested in the study topic and the sample size was only 20 women. The representativeness and potential selection bias is one of the major problems of the study.

2. In table1, “education level” was not defined in methods.

3. The results of each theme were not concise, and some other interpretations might also possible, which should be discussed in full.

4. There are some published articles that investigated views of women with prior GDM on about diet and physical activity interventions. A detailed discussion about the similarity and differences and the underlying reasons is warranted.

6. PLOS authors have the option to publish the peer review history of their article (what does this mean?). If published, this will include your full peer review and any attached files.

Reviewer #1: No

Reviewer #2: No

- - - - -

---

## [Author Response · Author response to Decision Letter 0]

14 Apr 2021

Please see the cover letter for our response to the reviewers' feedback (with clearer formatting). I have copied this below for convenience.

_____

The Primary Care Unit

Department of Public Health and Primary Care

University of Cambridge

Cambridge, UK

CB2 0SR

Prof Wing Hung Tam

Academic Editor

PLOS ONE

14 April 2021

PONE-D-21-01829

“Post-GDM support would be really good for mothers”: a qualitative interview study exploring how to support a healthy diet and physical activity after gestational diabetes

Dear Prof Tam

Thank you for considering our manuscript and for the opportunity to resubmit it. We would also like to thank the reviewers for their time and feedback on our study.

We have concerns about one of the criticisms of this study. As requested, please see a point-by-point response to each of the comments in the decision letter below. We have also marked the resulting changes in the manuscript.

We look forward to hearing from you and hope that this manuscript can be considered for publication in PLoS ONE.

Yours sincerely,

Rebecca Dennison, Simon Griffin, Juliet Usher-Smith, Rachel Fox, Catherine Aiken and Claire Meek 

*** 

Decision letter (email)

Dear Dr. Dennison,

Thank you for submitting your manuscript to PLOS ONE. After careful consideration, we have decided that your manuscript does not meet our criteria for publication and must therefore be rejected.

I am sorry that we cannot be more positive on this occasion, but hope that you appreciate the reasons for this decision.

Yours sincerely,

Wing Hung Tam

Academic Editor

PLOS ONE

***

Additional Editor Comments (if provided)

Thank you for the submission to Plos One. Given the reviewers comment and the small number of highly selected participants on a qualitative review, the manuscript is not considered to be suitable for publication in Plos One.

RESPONSE:

Thank you for considering our manuscript. We would also like to thank the reviewers for their time and feedback on our study.

Reviewer 1 reported that the manuscript is technically sound and had six suggestions for improving it. These points are simple for us to address.

The decision to reject the manuscript appears to be based on Reviewer 2’s feedback – primarily ‘the small number of highly selected participants on a qualitative’ study, therefore concerns over representativeness and potential selection bias. We would like to briefly raise the following points:

1. In qualitative research, we are not interested in generalisability but understanding a phenomenon of interest in depth [1] – in this case, this group of women with a history of GDM’s views on support for postpartum behaviour change.

2. We planned the sample size using the concept of information power [2], which we have added to the Methods of the manuscript (lines 87-90 of the highlighted manuscript): “We planned to interview approximately 20 women in order to reach data saturation, based on the relatively low information power anticipated [2]. This was because this study had a broad aim, sparse sample specificity but used purposive sampling and was a cross-case analysis. The interviews were structured around pre-defined recommendations.” As explained in the manuscript, we stopped interviewing when the data collection process no longer offers any new or relevant data (data saturation) [3].

3. Our sample size of 20 participants is larger than many semi-structured interview studies. For example, the qualitative systematic review that informed this interview study included 12 studies using face-to-face interviews [4]. Only three of those had a sample size >20 participants (23, 23 and 35 participants) and median sample size was 16 participants. PLoS ONE has also published several qualitative interview studies with a comparable number of participants in recent months (e.g. [5,6]). 

***

Reviewers' comments

***

Reviewer's Responses to Questions

1. Is the manuscript technically sound, and do the data support the conclusions?

Reviewer #1: Yes

Reviewer #2: Partly

2. Has the statistical analysis been performed appropriately and rigorously? 

Reviewer #1: Yes

Reviewer #2: No

RESPONSE: Note, there is no quantitative/statistical analysis in this qualitative study.

3. Have the authors made all data underlying the findings in their manuscript fully available?

Reviewer #1: No

Reviewer #2: No

RESPONSE: Note, the data underlying the findings of this study are interview transcripts. We have ethical approval and consent from participants to share the pseudo-anonymised transcripts with researchers upon their request to us directly, but not to make them publicly available. 

As suggested in the PLoS ONE Editorial policies for availability of qualitative data, we have made excerpts of the transcripts available within the paper and upon request.

We will include a data availability statement such as that of Backhausen et al. or Fleming et al. [5,6] (qualitative studies published in PLoS ONE in 2021).

4. Is the manuscript presented in an intelligible fashion and written in standard English?

Reviewer #1: Yes

Reviewer #2: Yes

5. Review Comments to the Author

Please use the space provided to explain your answers to the questions above. You may also include additional comments for the author, including concerns about dual publication, research ethics, or publication ethics. (Please upload your review as an attachment if it exceeds 20,000 characters).

***

Reviewer #1: Thank you for the opportunity to review this manuscript. This study aims to conduct a qualitative interview study on the views of women with history of GDM on the need for post-partum support. Given the high prevalence of GDM (affecting 10-20% of all pregnancies in many parts of the world), and the high risk of progressing to T2D after GDM (approximately 7-8x risk compared to non-GDM women), locally-relevant strategies to support women with GDM to reduce progression to T2D are very important. This study provides a systematic assessment of the attitudes and needs of women with GDM, and provide some useful information, in particular towards developing interventions and support networks within the UK healthcare setting and beyond.

RESPONSE: We thank the reviewer for reviewing our paper and for reinforcing the importance of these findings for intervention development. 

Major comments

Introduction

More reference to the risk of T2DM after GDM might be helpful in the introduction. This may include data from systematic review highlighting the increased risk of T2D after GDM compared to non-GDM pregnancies.

RESPONSE: We thank the reviewer for this suggestion. We have expanded the description of T2D risk after GDM with data from recent systematic reviews including absolute risk data, relative risk data and factors associated with higher rates of progression: “GDM is associated with increased risk of pregnancy complications in both mother and baby, and maternal cardiometabolic disorders in later life [7]. Approximately a third of women with GDM are diagnosed with type 2 diabetes (T2D) by 15 years postpartum, with recent data suggesting that the increased risk is sustained over time since GDM rather than being limited to the first few years after delivery [8]. T2D risk factors including high body mass index (BMI) and ethnicity further increase T2D risk in women who have had GDM: development of T2D is 18% (5–34%) higher per unit BMI at follow-up, and 57% (39–70%) lower in White European populations compared to other populations (adjusting for ethnicity and follow-up) [8]. Factors such as poorer pregnancy glucose tolerance requiring treatment with insulin have been found to further increase risk [9]. Overall, women who had GDM are 7–10 times more likely to develop T2D over their lifetime than women with normoglycaemic pregnancies [8,10,11].” (lines 48-59).

Page. 4 Line 82

It was stated that all women had history of GDM. Please clarify the prevailing diagnostic criteria which would have been used to diagnose GDM in these subjects.

RESPONSE: We have expanded this section to describe the NICE guidelines for diagnosing GDM at this time, plus some details on GDM management (lines 98-109): “NICE recommends screening for GDM with a 75g 2 hour oral glucose tolerance test (OGTT) in women with one or more risk factors (BMI greater than 30 kg/m2, previous baby weighing 4.5 kg or more, previous pregnancy affected by GDM, family history of diabetes, and ethnicity with a high prevalence of diabetes) [12]. Diagnostic cut-offs were defined according to local protocols: at Peterborough Hospital, those with a fasting value ≥5.6 mmol/l or 2 hour value of ≥7.8 mmol/l were diagnosed with GDM (NICE guidelines); at the Rosie Hospital, those with a fasting value ≥5.1 mmol/l, 1 hour value of ≥10.0 mmol/l or 2 hour value of ≥8.5 mmol/l were diagnosed with GDM (International Association of Diabetes in Pregnancy Study Groups (IADPSG) criteria [13]). Screening usually takes place at 24 to 28 weeks gestation, although can be repeated if the clinicians suspect GDM has developed. Following GDM diagnosis, women are closely managed with the aim of reducing glycaemia. This involves blood glucose monitoring, diet and exercise, and sometimes insulin and metformin medication.”)

Page 6 Line 135

It was stated that none of the participants had been diagnosed with T2D. Please clarify if all the women with GDM had undergone postpartum OGTT screening.

RESPONSE: We thank the reviewer for raising this point – none of the participants had been diagnosed but not all have been screened for diabetes. We have revised this to read “None of the 16 participants who had had a diabetes screening test since pregnancy had been diagnosed with T2D.” (lines 158-159).

Table 1

Is any information available on the proportion of women who were overweight/obese, or their mean BMI?

RESPONSE: This is a valid point to raise, given the association between postpartum BMI and diabetes risk explained in the introduction. However, we did not collect this data since we did not access patients’ medical records and did not ask them to self-report BMI in the demographics questionnaire at the end of the interview. We considered that this would be an insensitive question to ask in the context of this interview and during the postpartum period, following a pregnancy where many women are particularly sensitive about weight (some participants noted the stigma associated with GDM, particularly for overweight women, and how this could lead them to disengage with healthcare/behaviour change).

Furthermore, it is important for all women to maintain a healthy diet and be physically active after GDM, regardless of their weight/BMI.

We have commented on this in the Strengths and Limitations section in order to bring it to the readers’ attention: “We did not collect data on BMI or whether participants were overweight because we did not access their medical record, and to ask them to self-report this could be insensitive during the postpartum period. It is important for all women to maintain a healthy diet and be physically active after GDM, regardless of their BMI, yet we were unable to comment on whether BMI affected requirements for support.” (lines 353-357).

Page 8 Line 151

Although I understand this is a qualitative interview study, are the authors able to state the number of women who were aware of the link between GDM and T2D, which would significantly affect the interpretation of the results?

RESPONSE: Sixteen of the 20 participants were clearly aware of the association between GDM and T2D. Two participants were not aware of the association and two participants seemed to be inconsistent across their interviews.

We have added this information to this paragraph (lines 173-177), stating, “The remaining participants reported sentiments such as “I don’t feel like I've been given the help that I think there should be really out there” [Participant 1, attempting healthier postpartum lifestyle but felt unsupported overall], “post-GDM support would be really good for mothers” [P2, healthier, unsupported] and two participants explained that they had been unaware of an association between GDM and T2D.”

It is also worth noting that the interview was consistently framed in the context of preventing T2D in women who had had GDM (e.g. this was the first sentence of the participant information sheet and explained in the introduction to the interview). We suggested that the participants seek advice from their GP if they had questions or concerns.

Page 14 Line 246-248

Please provide further details on the responses from the participants on the preferred format, source and timing of providing support. Are the responses summarized by the later comments on In person peer groups, appointments with healthcare professionals and written messages?

RESPONSE: Yes, the participants’ responses on the preferred format, source and timing of support are reported in the section “Delivery of support or interventions” (lines 271-311). We have clarified this by saying “This included the preferred format (including in-person peer support groups, appointments with healthcare professionals and written information), source and timing.” (lines 272-274).

***

Reviewer #2: The authors conducted semi-structured interviews with 20 participants with a history of GDM to explored the views of women on possible interventions to support healthy diet and physical activity to reduce diabetes risk and aimed to identify the most promising interventions for future development. The sample size was small and less representative. I also have some other concerns about the interpretations and discussions.

Major comments:

1. The participants were women who were interested in the study topic and the sample size was only 20 women. The representativeness and potential selection bias is one of the major problems of the study.

RESPONSE: As we describe in our response to the editorial comments above, we would like to respond with the following brief points:

1. In qualitative research, we are not interested in generalisability but understanding a phenomenon of interest in depth [1] – in this case, this group of women with a history of GDM’s views on support for postpartum behaviour change.

2. We planned the sample size using the concept of information power [2], which we have added to the Methods of the manuscript (lines 87-90): “We planned to interview approximately 20 women in order to reach data saturation, based on the relatively low information power anticipated [2]. This was because this study had a broad aim, sparse sample specificity but used purposive sampling and was a cross-case analysis. The interviews were structured around pre-defined recommendations.” As explained in the manuscript, we stopped interviewing when ‘the data collection process no longer offers any new or relevant data’ (data saturation) [3].

3. Our sample size of 20 participants is larger than many semi-structured interview studies. For example, the qualitative systematic review that informed this interview study included 12 studies using face-to-face interviews [4]. Only three of those had a sample size >20 participants (23, 23 and 35 participants) and median sample size was 16 participants. PLoS ONE has also published several qualitative interview studies with a comparable number of participants in recent months (e.g. [5,6]).

In addition, recruiting participants who are interested in the study is a universal source of bias in every research study. We have already commented on this in the Strengths and Limitations section: “This [the need for postpartum support] may have been influenced by recruitment bias, with more health-conscious women or those in need of particular support more likely to engage in the study.” (lines 352-353).

2. In table1, “education level” was not defined in methods.

RESPONSE: The different education levels reported in Table 1 are universal terms to describe education in the UK. Nevertheless, we will include examples of qualifications gained at these levels in Table 1 (under line 162): 

• Secondary or further (GCSEs, A levels, BTEC, apprenticeships or equivalent)

• Higher (Bachelor’s degree or equivalent)

• Postgraduate (Master’s degree, PhD or equivalent)

3. The results of each theme were not concise, and some other interpretations might also possible, which should be discussed in full.

RESPONSE: We are not clear from this comment which of the Results are not concise. Each of the themes is summarised in one to three short paragraphs, with a total word count that is shorter than many quantitative papers.

As is true for all qualitative research, we acknowledge that other interpretations of these data might also be possible, and have acknowledged this in the Strengths and Limitations section by concluding with “Finally, as is true for all qualitative research, other interpretations of the data collected in this study might be possible, although no participants disagreed with the summary of the findings that we sent to them.” (lines 360-362). We came to these conclusions as a multidisciplinary team with expertise in qualitative research, GDM healthcare and primary care after being immersed in the interview data. We have tried to be as transparent as possible in presenting the findings, exploring deviant cases and presenting evidence (e.g. quotations) so that the reader is able to draw their own conclusions. We have also added the coding frame to the supporting information to increase transparency (line 146: “The final codebook for the framework is reported in S2 Table.”).

There must always be a balance between being concise and reporting in enough detail to explain how we came to these interpretations.

4. There are some published articles that investigated views of women with prior GDM on about diet and physical activity interventions. A detailed discussion about the similarity and differences and the underlying reasons is warranted.

RESPONSE: In lines 321-332, we compared the findings of this study to our recent systematic review on women’s views towards having a healthy diet and physical activity after GDM (similarities, differences and underlying reasons) [4]. This review includes intervention studies as well as those investigating barriers and facilitators to the behaviours (e.g. Nicklas et al. 2011, O’Dea et al. 2015). We have expanded and clarified this section to read: “… We found that influences on healthy diet and exercise were similar between the review and this interview study, such as spending time with children instead of exercising and how the family could facilitate healthy behaviours. Lack of time and energy were particularly evident in the early postpartum period in this interview study, which was considered to be a time for learning to adapt to life with their new baby. Women were more supportive of integrating activity into their daily routine than of participating in family-based exercise activities, which appeared to be important in the review. …”

Additionally, in lines 333-343, we compared our findings on timing of the intervention to those of previous studies related to this topic. We have added reference to Ingstrup et al. 2019 (nine women’s experience with peer counselling and social support during a lifestyle intervention among women with previous GDM), which highlighted that rapport was important for social support to be effective: “We concluded that women with GDM should be prepared for more specific follow-up interventions during their pregnancy, provided that this is done in a sensitive manner, echoing the findings of Ingstrup et al. regarding the importance of rapport with peer councillors [14].” (lines 339-342).

We would be happy to add any other recent papers the reviewer is aware of. 

***

References

1. Myers M. Qualitative Research and the Generalizability Question: Standing Firm with Proteus. Qual Rep. 2000; Available from: https://nsuworks.nova.edu/tqr/vol4/iss3/9/. Date last accessed: 01 March 2021.

2. Malterud K, Siersma VD, Guassora AD. Sample size in qualitative interview studies. Qual Health Res. 2016;26(13):1753–60. 

3. Dworkin SL. Sample Size Policy for Qualitative Studies Using In-Depth Interviews. Arch Sex Behav. 2012;41(6):1319–20. 

4. Dennison RA, Ward RJ, Griffin SJ, Usher-Smith JA. Women’s views on lifestyle changes to reduce the risk of developing Type 2 diabetes after gestational diabetes: a systematic review, qualitative synthesis and recommendations for practice. Diabet Med. 2019;36(6):702–17. 

5. Fleming T, Collins AB, Bardwell G, Fowler A, Boyd J, Milloy MJ, et al. A qualitative investigation of HIV treatment dispensing models and impacts on adherence among people living with HIV who use drugs. Ahmed SI, editor. PLoS One. 2021;16(2):e0246999. 

6. Backhausen MG, Iversen ML, Sköld MB, Thomsen TG, Begtrup LM. Experiences managing pregnant hospital staff members using an active management policy—A qualitative study. Doraiswamy S, editor. PLoS One. 2021;16(2):e0247547. 

7. Okoth K, Chandan JS, Marshall T, Thangaratinam S, Thomas GN, Nirantharakumar K, et al. Association between the reproductive health of young women and cardiovascular disease in later life: umbrella review. BMJ. 2020;371:m3502. 

8. Dennison RA, Chen ES, Green ME, Legard C, Kotecha D, Farmer G, et al. The absolute and relative risk of type 2 diabetes after gestational diabetes: A systematic review and meta-analysis of 129 studies. Diabetes Res Clin Pract. 2021;171:108625. 

9. Rayanagoudar G, Hashi AA, Zamora J, Khan KS, Hitman GA, Thangaratinam S, et al. Quantification of the type 2 diabetes risk in women with gestational diabetes: A systematic review and meta-analysis of 95,750 women. Diabetologia. 2016;59(7):1403–11. 

10. Song C, Lyu Y, Li C, Liu P, Li J, Ma RC, et al. Long-term risk of diabetes in women at varying durations after gestational diabetes: a systematic review and meta-analysis with more than 2 million women. Obes Rev. 2018;19(3):421–9. 

11. Vounzoulaki E, Khunti K, Abner SC, Tan BK, Davies MJ, Gillies CL. Progression to type 2 diabetes in women with a known history of gestational diabetes: systematic review and meta-analysis. BMJ. 2020;369:m1361. 

12. National Institute for Health Care and Excellence. Diabetes in pregnancy: management from preconception to the postnatal period. NICE Clinical Guideline. 2015. Available from: www.nice.org.uk/guidance/ng3/chapter/introduction. Date last accessed: 01 March 2021. 

13. Metzger BE, International Association of Diabetes and Pregnancy Study Groups Consensus Panel, Metzger BE, Gabbe SG, Persson B, Buchanan TA, et al. International Association of Diabetes and Pregnancy Study Groups recommendations on the diagnosis and classification of hyperglycemia in pregnancy. Diabetes Care. 2010;33(3):676–82. 

14. Ingstrup MS, Wozniak LA, Mathe N, Butalia S, Davenport MH, Johnson JA, et al. Women’s experience with peer counselling and social support during a lifestyle intervention among women with a previous gestational diabetes pregnancy. Heal Psychol Behav Med. 2019;7(1):147–59.

---

## [Decision Letter · Decision Letter 1]

10 Aug 2021

PONE-D-21-01829R1

“Post-GDM support would be really good for mothers”: a qualitative interview study exploring how to support a healthy diet and physical activity after gestational diabetes

PLOS ONE

Dear Dr. Dennison,

Thank you for submitting your manuscript to PLOS ONE. After careful consideration, we feel that it has merit but does not fully meet PLOS ONE’s publication criteria as it currently stands. Therefore, we invite you to submit a revised version of the manuscript that addresses the points raised during the review process.

We look forward to receiving your revised manuscript.

Kind regards,

Or Kan Soh

Academic Editor

PLOS ONE

Journal Requirements:

Additional Editor Comments (if provided):

Dear Author

Based on the feedback from the reviewers, you are subjected to major revisions. You must strictly adhere to the comments rendered to you.

Thank you.

Reviewers' comments:

Reviewer's Responses to Questions

**Comments to the Author**

1. If the authors have adequately addressed your comments raised in a previous round of review and you feel that this manuscript is now acceptable for publication, you may indicate that here to bypass the “Comments to the Author” section, enter your conflict of interest statement in the “Confidential to Editor” section, and submit your "Accept" recommendation.

Reviewer #3: (No Response)

Reviewer #4: (No Response)

Reviewer #5: (No Response)

Reviewer #6: All comments have been addressed

2. Is the manuscript technically sound, and do the data support the conclusions?

Reviewer #3: Partly

Reviewer #4: Partly

Reviewer #5: Partly

Reviewer #6: Yes

3. Has the statistical analysis been performed appropriately and rigorously? 

Reviewer #3: Yes

Reviewer #4: N/A

Reviewer #5: N/A

Reviewer #6: Yes

4. Have the authors made all data underlying the findings in their manuscript fully available?

Reviewer #3: Yes

Reviewer #4: No

Reviewer #5: Yes

Reviewer #6: Yes

5. Is the manuscript presented in an intelligible fashion and written in standard English?

Reviewer #3: Yes

Reviewer #4: Yes

Reviewer #5: Yes

Reviewer #6: Yes

6. Review Comments to the Author

Reviewer #3: FPG test ?? in Figure 2. Please provide the full form of FPG test before providing the abbreviated form.

Reviewer #4: Thank you very much for having me to review the work entitled “Post-GDM support would be really 1 good for mothers”: a qualitative interview study exploring how to support a healthy diet and physical activity after gestational diabetes.

The study aims to explore women’s views on suggested practical approaches to achieve and maintain a healthy diet and physical activity to reduce T2D risk. Please consider the following comments to improve the manuscript.

1. Abstract

Line 33 – Line 35: Suggest to include the percentages of “a third” participants for “transformative”, “beneficial” and “did not want additional support”.

Line 35: Suggest to put the percentage after “the majority”.

Line 38: Suggest to put the percentage after “four”.

2. Introduction

Line 55 - Line 56: Since you mentioned “These sites were chosen to provide socioeconomic and ethnic diversity, and represent views from those attending both secondary and tertiary centres offering GDM/ and obstetric care” in line 80 to line 81 (page 4), hence, it is better to specify the prevalence of GDM of different ethnicities (Chinese, Indian, Japanese) instead of putting “as compare to other populations”.

Line 57: Please specify the type of risk for “glucose tolerance requiring treatment with insulin have been found to further increase risk”. It this refer to GDM risk or T2D risk?

Line 64: Please specify the activities of behaviour change intervention. For example, the type of intervention done by previous researchers to give a better picture for the readers. Support your sentence with reference.

Line 65: Please specify the type of population. Is this referring to women with GDM history or without GDM history? Because GDM history may influence the effectiveness of the behavioural change intervention.

Line 66 - Line 67: Please provide a reference for “we found that women who had had GDM identified themselves primarily as mothers who prioritised their family above themselves”.

Line 71: Considering a small sample size in the present study, please provide the sample size for Reference 13 and Reference 14. This is to give an overall picture for the readers to understand the common sample size being used in qualitative research.

Line 71: Please provide brief details for Reference 13 and Reference 14. What are the previous activities conducted by the researchers?

Line 72: Please mention the research gap before your study adjective.

Line 74: Is this objective “We aimed to identify the most promising interventions for future development” reflect in your findings? If not, please remove this sentence.

3. Recruitment

Line 86: Please provide the response rate of this study. Out of the total contacted participants, how many of them agreed or rejected to participate in the present study.

Line 87 – Line 89: These sentences are not strong enough to support your final sample size (n=20). Since you mentioned “to reach data saturation”, have you conducted data saturation assessment in your study? Please explain how you assess the data saturation.

Line 88 – Line 89: What do you mean by “this study had a broad aim sparse sample specificity but used purposive sampling and was a cross-case analysis”? This sentence is unclear and needs further improvement.

Line 90: What do you mean by “the interviews were structured around pre-defined recommendation”? Please provide more details and also a reference for this statement.

Line 90 - Line 91: Is Reference 16 correctly cited?

Line 91 – Line 92: How many participants you have interviewed in your study? Do you include data of all participants that have completed the interview or only part of them? Please provide the response rate.

4. Inclusion criteria

Line 98: Do you conduct GDM screening in your study? Or do you obtain GDM data from the participants’ medical history?

Line 110 – Line 113: Please mentioned how many women you have excluded from the study.

5. Interview process

Do you conduct any pre-test prior to the interview? Modification of interview guide and suggestion cards are required before you conduct the actual interview. If yes, how many attended the pre-test and what modifications you have done?

Line 122 - Line 123: Based on Table S2, it seems like No 11 to 20 in Table S1 are not including in your study. Please clarify the number of suggestion cards in your study. Please mention how many suggestion cards you have included in the final version.

Line 132: Please elaborate type of format for “what format might be most effective”.

Line 133: Please mention how many suggestion cards you have shown to your participants.

Line 137: “These questions were then repeated for attending diabetes screening (reported separately)”: When did you conduct the diabetes screening? On the same day of the semi-structured interview? Do you conduct the diabetes screening on your own? Do you include the feedbacks of participants in your manuscript? Please provide the feedbacks of participants provided in the diabetes screening in the results or appendix.

Line 137 - Line 138: Do you collect social-demographic data or demographic data? Please specify the type of information you have collected for demographics (ethnicity, age, occupation and etc).

Line 138: Please write the full term of “RD” instead of its acronym.

6. Analysis

Line 142: Please provide brief details of the framework approach.

Line 143: Please provide the version number and manufacturer details of NVivo 12.

Line 143 – Line 144: “…and developed a thematic framework”. Please attach the thematic framework to the manuscript.

Line 144 – Line 145: Do you refine your thematic framework based on the first few interviews? Does the thematic framework represent the views of all participated participants?

Line 147 – Line 148: Please write the full term of “RD” and “RF” instead of their acronym.

Line 148: “… and charted four transcripts to ensure agreement”. Please explain how to assess agreement in your study. Do you have any references for the classification?

7. Results

Table 1: Please provide mean years of postpartum.

Table 1: Since this is a descriptive analysis, it will be good to present the number of participants for each ethnicity (Chinese, Japanese, Indian).

Table 1: Please check the distribution of participants of employment and maternity leave as their final number is more than 20.

Table 1: Please add the number of respondents who did not live with their partner.

Table 1: “……. All pregnancies affected by GDM”: Only 13 participants? Based on your inclusion criteria, you only include those with GDM history in your study. It is a confusing statement and please provide your justification.

Table 1: You mentioned in the methodology (Line 111 – Line 113) that “ …. Participated in a pregnancy or GDM-related intervention or were considered unsuitable… were not invited”. Please explain why you include those on medication of GDM as depicted in Table 1. Do you accept those with medication of GDM in your study?

Line 180: Please include the number of suggestion cards you have shown to the participants.

Line 187: How do you define the agreement? What are the cut-offs for overall agreement, overall mixed response and overall disagreement? Do you have any references for the classification?

Line 187: “… not all participants were shown each card, and some did not comment or agreement was unclear.”. Please justify why you exclude certain suggestion cards and how to define that “agreement was unclear”. This is important as it shows the quality of each suggestion card that you have proposed in your study.

Information and understanding

Line 193 – Line 194 and Line 200: “… most of the participants”; “some would add…” and “… others had poor awareness” and “since most already had general…”. Please provide the actual number of participants.

Improving diet

Line 203 and Line 206 – Line 207: “The majority of the …”; “A couple of participants…” and “Others wanted advice that was relevant…”. Please provide the actual number of participants.

Improving physical activity

Line 215, Line 217, Line 221, Line 224, Line 226 and Line 231: “… Although many participants…”; “Some preferred…”; “… a few...”; “Almost all the participants..” and “Several participants….”: Please provide the actual number of participants.

Family

Line 237, Line 241, Line 243 and line 245: “…. Others walked with …..”; Some also found ….”; “Some participant reasoned that….”; “Others already knew …..”; “those that agreed wanted ….”: Please provide the actual number of participants.

Money

Line 252 and Line 253: “other participants considered …..”; “Some noted that cooking…”: Please provide the actual number of participants.

Monitoring

Line 256 and Line 262: “Almost all of the participants ….”; “at the same time, several were cautions ….”: Please provide the actual number of participants.

Sustainability

Line 266: “The majority of the participants ….”: Please provide the actual number of participants.

What about others who did not want advice about sustaining changes? Please give more details on this.

Delivery of support or intervention

Line 272: “The participants also suggested ….”: How many of them give additional suggestions?

Appointment with healthcare professionals

Line 283 and Line 293: “Most participants were keen ……”; “Many suggested that …..”: Please provide the actual number of participants.

Written information

Line 300: “The participants thought that ……”: How many participants?

8. Discussion

Line 316: “… would help them to reduce their risk”: Please add “T2D” after the “risk”.

Line 317: “Many wanted more specific information about their long-term ……..”: Please add brief details of the specific information requested by the participants.

Line 323 – Line 324: “…. exercise between the review and this interview study”: Please give specific findings related to the previous review. Please add the reference of the review.

Line 326 – Line 327: “Lack of time and energy………, which was considered to be a time ……. new baby.”: Sentence unclear and do you have any reference to support this sentence? Please provide a reference to support this sentence.

Line 331: “…… many felt that more specific information about lifestyle……”: I believe that participants did provide specific information that they want to know more in the qualitative interview. Please add in the what are the specific information requested by the participants.

Line 334: “….. tended only to maintain selected elements of the GDM diet because…”: Please elaborate on the elements or example of the GDM diet so that readers know how does GDM diet looks like. If GDM diet is difficult to sustain, then what mothers can do? Any other diet that is suitable for the mothers to tackle T2D?

Line 335: “Interventions may therefore support…”: Please suggest the type of intervention that will be beneficial to the participants.

Line 336: “ …. these changes were anticipated help women ….”: The “changes” here refer to what kind of change? Please specify the type of change in your text.

Line 338 – Line 339: Please briefly explain the previous findings of previous research.

Line 340: “for more specific follow-up interventions during their pregnancy…:” Please suggest a suitable specific follow-up intervention in your conclusion.

9. Strength and limitations

Line 357: “yet we were unable …… for support”: Do you think future study needs to add in BMI data in the interview to elucidate the influence of BMI towards the requirement for support? If yes, please add a sentence on this.

10. Implications for practice

Line 372: “support the important role clinicians play…”: Please add “of” before “clinicians”.

11. Supporting information

Table S3: How do you define the agreement? What are the cut-offs for overall agreement, overall mixed response and overall disagreement? Do you have any references for the classification?

Reviewer #5: In the methods section, the trustworthiness of the data was not indicated.

Data collection technique analysis and report were not meticulously stated. The author should have used COREQ.

The authors have not clearly stated how the themes emerged. They have used a card which is pre-specified and may not explore the phenomenon very well

The quotations used in your manuscript contain potentially identifying information. Please amend your manuscript by either limiting the amount of potentially-identifying information presented, or by obtaining explicit consent to publish such information. To limit the identifying information, please remove the ages and occupation information from your quotes to help maintain participant confidentiality. Please use age ranges in place of the ages. Please check that the identifiers do not link to participants.

As far as qualitative research is concerned, descripting in number is not recommended. How ever, the authors stated state

the number of women who were aware of the link between GDM and T2D, which would significantly affect the interpretation of the results.

Generally, the title is of interest and current issue.

The paper can be accepted for publication after modification.

Reviewer #6: (No Response)

7. PLOS authors have the option to publish the peer review history of their article (what does this mean?). If published, this will include your full peer review and any attached files.

Reviewer #3: No

Reviewer #4: No

Reviewer #5: No

Reviewer #6: No

---

## [Author Response · Author response to Decision Letter 1]

28 Sep 2021

Please see the cover letter for our response to the reviewers' feedback.

Many thanks.

---

## [Decision Letter · Decision Letter 2]

20 Dec 2021

PONE-D-21-01829R2“Post-GDM support would be really good for mothers”: a qualitative interview study exploring how to support a healthy diet and physical activity after gestational diabetesPLOS ONE

Dear Dr. Dennison,

Thank you for submitting your manuscript to PLOS ONE. After careful consideration, we feel that it has merit but does not fully meet PLOS ONE’s publication criteria as it currently stands. Therefore, we invite you to submit a revised version of the manuscript that addresses the points raised during the review process.

We look forward to receiving your revised manuscript.

Kind regards,

Diane Farrar

Academic Editor

PLOS ONE

Journal Requirements:

Reviewers' comments:

Reviewer's Responses to Questions

**Comments to the Author**

1. If the authors have adequately addressed your comments raised in a previous round of review and you feel that this manuscript is now acceptable for publication, you may indicate that here to bypass the “Comments to the Author” section, enter your conflict of interest statement in the “Confidential to Editor” section, and submit your "Accept" recommendation.

Reviewer #4: All comments have been addressed

Reviewer #5: All comments have been addressed

2. Is the manuscript technically sound, and do the data support the conclusions?

Reviewer #4: Yes

Reviewer #5: (No Response)

3. Has the statistical analysis been performed appropriately and rigorously? 

Reviewer #4: Yes

Reviewer #5: (No Response)

4. Have the authors made all data underlying the findings in their manuscript fully available?

Reviewer #4: No

Reviewer #5: (No Response)

5. Is the manuscript presented in an intelligible fashion and written in standard English?

Reviewer #4: Yes

Reviewer #5: (No Response)

6. Review Comments to the Author

Reviewer #4: Previous comments were addressed by the authors. Please confirm the distribution of respondents under the maternity leave in Table 1 is correctly presented as the current sample size is less than 20.

The paper is ready to be accepted after minor modification.

Reviewer #5: I appreciate the authors' efforts. This revised draft addresses all of my concerns. So I don't have any more comments or questions.

7. PLOS authors have the option to publish the peer review history of their article (what does this mean?). If published, this will include your full peer review and any attached files.

Reviewer #4: No

Reviewer #5: No

---

## [Author Response · Author response to Decision Letter 2]

4 Jan 2022

Dear Dr Farrar

Thank you for considering our manuscript and for the opportunity to resubmit it. We would also like to thank the reviewers for their time and feedback on our study.

We have addressed the outstanding points (Review Comments to the Author) as follows:

> Reviewer #4: Previous comments were addressed by the authors. Please confirm the distribution of respondents under the maternity leave in Table 1 is correctly presented as the current sample size is less than 20. The paper is ready to be accepted after minor modification.

> RESPONSE: The distribution of respondents on maternity leave in Table 1 is correct. One participant was a ‘home parent’ therefore they cannot be on maternity leave from formal employment. We have added ‘NA’ in order to clarify this, with the changes highlighted.

> Reviewer #5: I appreciate the authors’ efforts. This revised draft addresses all of my concerns. So I don't have any more comments or questions.

There were no other comments, and all of the references are complete and correct.

We look forward to hearing from you and hope that this manuscript can be accepted for publication in PLoS ONE.

Yours sincerely,

Rebecca Dennison, Simon Griffin, Juliet Usher-Smith, Rachel Fox, Catherine Aiken and Claire Meek

---

## [Editor Report · Decision Letter 3]

7 Jan 2022

“Post-GDM support would be really good for mothers”: a qualitative interview study exploring how to support a healthy diet and physical activity after gestational diabetes

PONE-D-21-01829R3

Dear Dr. Dennison,

We’re pleased to inform you that your manuscript has been judged scientifically suitable for publication and will be formally accepted for publication once it meets all outstanding technical requirements.

Kind regards,

Diane Farrar

Academic Editor

PLOS ONE

---

## [Editor Report · Acceptance letter]

11 Jan 2022

PONE-D-21-01829R3 

“Post-GDM support would be really good for mothers”: a qualitative interview study exploring how to support a healthy diet and physical activity after gestational diabetes 

Dear Dr. Dennison:

I'm pleased to inform you that your manuscript has been deemed suitable for publication in PLOS ONE. Congratulations! Your manuscript is now with our production department. 

Kind regards, 

on behalf of

Dr. Diane Farrar 

Academic Editor

PLOS ONE